# Genome-Wide Identification and Analysis of OsSPXs Revealed Its Genetic Influence on Cold Tolerance of Dongxiang Wild Rice (DXWR)

**DOI:** 10.3390/ijms24108755

**Published:** 2023-05-15

**Authors:** Cheng Huang, Jilin Wang, Dianwen Wang, Jingjing Chang, Hongping Chen, Dazhou Chen, Wei Deng, Chunjie Tian

**Affiliations:** 1Rice National Engineering Research Center (Nanchang), Jiangxi Academy of Agricultural Sciences, Nanchang 330200, China; chenghuang@webmail.hzau.edu.cn (C.H.); wangjilin1982@163.com (J.W.); dianwen1989@126.com (D.W.); 13970920363@139.com (H.C.); cdz288@163.com (D.C.); dw19710330@163.com (W.D.); 2Key Laboratory of Mollisols Agroecology, Northeast Institute of Geography and Agroecology, Chinese Academy of Sciences, Changchun 130102, China; changjingjing@iga.ac.cn

**Keywords:** Dongxiang wild rice, *Oryza rufipogon* Griff., SPX-domain protein, cold tolerance, expression patterns

## Abstract

SPX-domain proteins (small proteins with only the SPX domain) have been proven to be involved in phosphate-related signal transduction and regulation pathways. Except for *OsSPX1* research showing that it plays a role in the process of rice adaptation to cold stress, the potential functions of other *SPX* genes in cold stress are unknown. Therefore, in this study, we identified six *OsSPXs* from the whole genome of DXWR. The phylogeny of *OsSPXs* has a strong correlation with its motif. Transcriptome data analysis showed that *OsSPXs* were highly sensitive to cold stress, and real-time PCR verified that the levels of *OsSPX1*, *OsSPX2*, *OsSPX4*, and *OsSPX6* in cold-tolerant materials (DXWR) during cold treatment were higher than that of cold-sensitive rice (GZX49). The promoter region of DXWR *OsSPXs* contains a large number of cis-acting elements related to abiotic stress tolerance and plant hormone response. At the same time, these genes have expression patterns that are highly similar to cold-tolerance genes. This study provides useful information about *OsSPXs*, which is helpful for the gene-function research of DXWR and genetic improvements during breeding.

## 1. Introduction

Phosphorus (P) is essential for plant growth and development, and inorganic phosphate (Pi) is a form of phosphorus available to plants [1,2,3]. SPX-domain proteins are widely found in eukaryotes, including animals, plants, and fungi. This domain is named according to the first letter of the three proteins Syg1, Pho81, and XPRI [4,5]. SPX-domain proteins have been proven to participate in phosphate-related signal transduction and regulation pathways. In plants, SPX has four subfamilies, namely, SPX-EXS (named after Saccharomyces cerevisiae, Erd1; mammalian, Xpr1; and Saccharomyces cerevisiae, Syg1), SPX-MFS (a major promoter superfamily), SPX ring (a really interesting new gene), and SPXs (small proteins of the SPX domain) [6,7,8]. In Arabidopsis, SPX-EXS proteins (such as AtPHO1) are involved in transferring Pi from roots to buds [9]. The *pho1* mutant is characterized by a serious lack of Pi in stems, but the content of Pi in roots is normal [9,10]. In addition, three members of the *AtPHO1* family showed possible interactions with signaling pathways involved in Pi deficiency and responses to auxin, cytokinin, and abscisic acid (ABA) [11,12,13]. AtVPT1 and OsSPX-MFS1, belonging to SPX-MFS, can transfer Pi from the cytoplasm to vacuoles [14,15]. For proteins with only the SPX domain, four members were identified in Arabidopsis thaliana (*AtSPX1*—*AtSPX4*) and six members in rice, (*OsSPX1*—*OsSPX6*) [16]. When Pi is sufficient, SPX proteins, including OsSPX1, OsSPX2, OsSPX4, and OsSPX6, can also inhibit the activity of OsPHR2 in the nucleus by blocking OsPHR2 from binding to the PSI gene promoter in a phosphate-dependent manner. This prevents PHR2 from entering the nucleus, thereby inhibiting PHR2 from binding to cis-acting elements [17]. A similar mechanism has been found in Arabidopsis [18,19,20,21].In addition, *OsSPX1* also plays a role in cold resistance. Overexpression of *OsSPX1* can enhance the cold resistance of rice, Arabidopsis, and tobacco, and downregulation of *OsSPX1* expression leads to high sensitivity of rice seedlings to low temperatures and oxidative stress [22,23]. Moreover, low-concentration inorganic phosphate (Pi) has been proven to play an important role in the adaptation process of Arabidopsis thaliana to low temperatures [24]. However, the information on how *OsSPXs* respond to cold stress is still very limited.

Rice is one of the most important food crops in the world [25]. It is native to temperate climate regions and is very sensitive to low-temperature stress, which endangers global agricultural production [26,27,28]. It is important to cultivate stress-tolerant varieties that can resist the rising unstable temperatures caused by global climate change [29,30,31]. Cold stress can be divided into freezing stress (below 0 °C) or low-temperature stress (0–20 °C) according to different physiological reactions in the temperature range of cold injury [32]. Exploring the mechanism of regulating cold tolerance will help to promote the development of molecular breeding and improve the cold tolerance of rice. DXWR is a common wild rice and has the highest latitude and the most northerly distribution found so far. It is rich in disease- and insect-resistance genes, as well as submergence-, cold-, drought-, and barren-resistance genes [33,34]. DXWR is rich in cold-tolerance genes, has high tolerance to cold stress, and can safely survive a winter temperature of minus 12.8 °C [35]. Several QTLs and candidate genes related to cold tolerance have been identified in DXWR [36,37]. Therefore, it is of great significance to improve the cold adaptability of cultivated rice by identifying the cold-tolerance-related genes of DXWR. Up until now, genome data of DXWR have been published [38]; however, the *SPX* gene members of DXWR and their potential importance have not been studied. Therefore, this study comprehensively analyzed the sequence composition, gene structure, cis-acting elements, and gene-replication events of the six *OsSPXs* in the genome of DXWR. At the same time, the evolutionary relationships of *OsSPXs* among DXWR, maize, and wheat were analyzed. The transcriptome data of different tissues and low-temperature stress were used to understand the expression pattern of *OsSPXs* in DXWR. In addition, we identified differences between cold tolerant materials and cold-sensitive materials in *OsSPXs* using real-time PCR. This study laid a foundation for the functional research and genetic breeding of DXWR.

## 2. Results

### 2.1. Phylogenetic Tree Analysis and Motif Analysis of OsSPXs in DXWR

The protein data of DXWR OsSPXs (named OsSPX1~6 according to each gene’s corresponding gene in Nipponbare) were obtained through the ORYZA GENOME database (online: http://www.ricegermplasmgenome.com/, accessed on 26 March 2022). The OsSPXs development evolution tree was constructed through MEGA, and the online MEME server was used for motif analysis. OsSPX1 has the highest homology with OsSPX2; OsSPX3 has a close genetic relationship with OsSPX5, followed by OsSPX6; and OsSPX4 has a distance genetic relationship with other OsSPXs (Figure 1a). In combination with the motif analysis, motifs 1~5 exist in all OsSPXs (Figure 1b). Motifs 6 and 10 exist in OsSPX3 and OsSPX6, motif 7 is unique to OsSPX1 and OsSPX2, motif 8 only exists in OsSPX3 and OsSPX5, and motif 9 exists in all OsSPXs except OsSPX4 and OsSPX6 (Figure 1c).

### 2.2. Gene Collinearity Analysis of OsSPXs

Gene replication events are crucial to the evolution of gene families and often play an important role in gene amplification and the generation of new functional genes. Therefore, we have studied the repeat events of *OsSPXs* in the whole genome of DXWR and found that *OsSPXs* have no tandem repeat events. However, using BLASTP and MCScanX methods, three pairs of fragment-duplication events were found in *OsSPXs* (Figure 2a), and these fragment-duplication genes were distributed on chromosomes 2, 3, 6, 7, and 10. These results suggest that fragment-replication events play an important role in the amplification of *OsSPXs*.

We constructed a collinearity map of three gramineous plants (maize, DXWR, and wheat) (Figure 2b). Among the homologous *OsSPXs* of DXWR and other species, there were five pairs of genes homologous with maize and four pairs with wheat. Interestingly, *OsSPX1*, *4*, and *5* all have homologous genes in the three species. Due to polyploidization, some genes differentiate during evolution and generate new gene functions to adapt to the environment. The number of genes with collinear relationships in wheat is three times that of rice (*OsSPX1*, *4*, *5*, and *6*), indicating that hexaploid wheat with these genes existed before differentiation and played an important role in these wheat after differentiation.

### 2.3. Expression Patterns of OsSPXs in Different Rice Tissues and Their Responses to Cold Stress

The analysis of gene expression patterns can provide important information for identifying the biological functions of genes of interest. In order to study the expression pattern of *OsSPXs*, the expression data of *OsSPXs* stored in the public database were used to carry out such analyses on twelve tissues/organs of rice, including seeding root, young leaf, SAM, young inflorescence, inflorescence P2–P6, and seed S1–S5. As shown in Figure 3a, the expression spectrum thermogram shows that *OsSPX1*, *2*, and *4* are highly expressed at different growth stages. Other *OsSPXs* have tissue-specific expression patterns. For example, *OsSPX5* is only highly expressed in the seeding root. In addition, *OsSPX3* showed specific high expression at the late stage of seed development. However, *OsSPX6* only showed higher expression levels in mature leaves. These expression analyses showed that *OsSPX1*, *2*, and *4* genes may play a constitutive function in the process of plant growth and development, and that some genes, such as *OsSPX3*, *5*, and *6*, may play different roles in different tissues/organs and different seed-development stages. We used the transcriptome data of Dongxiang wild rice before and after treatment at 4 °C to further analyze the expression levels of *OsSPXs* under low-temperature stress. The results showed that, with the exception of *OsSPX3* and *OsSPX5*, the expression levels of *OsSPXs* did not change significantly after 4 h of treatment compared to 0 h of treatment; however, they increased more than two times after 12 h of treatment (Figure 3b). These *OsSPXs* may be involved in the cold-tolerance process of DXWR.

DXWR and GZX49 were treated for 14 days at 4 °C and then recovered at room temperature for 7 d. We counted the living plants (containing about 40% green leaves) and dead plants of the two materials to determine the survival rates. The survival rate of DXWR was 90%, whereas that of GZX49 was only 10% (Figure 3c). We believe that DXWR is a cold-resistant material and that GZX49 is cold-sensitive to cold stress. Then, real-time PCR (qPCR) was used to check their expressions after cold-stress (4 °C) treatment in seedlings of cold-tolerant wild rice (DXWR) and cold-sensitive indica rice (GZX49). Time expression profiles of four *OsSPXs* were obtained in DXWR and GZX49 under low-temperature stress (Figure 3d). In general, the four *OsSPXs* in both materials were induced due to cold stress. The relative expressions of the other three genes (*OsSPX1*, *2*, and *6*) reached their peaks at 24 h after treatment at 4 °C, with the exception of *OsSPX4*, which reached its maximum at 12 h after treatment at 4 °C. In addition, the expression levels of *OsSPX1* and *2* in DXWR were significantly higher than in GZX49 at 24 h after treatment at 4 °C, and the relative expression levels of *OsSPX4* and *6* in DXWR were significantly higher than in GZX49 after treatment at 4 °C at 4, 12, and 24 h. These results suggest that these four *OsSPXs* may play important roles in the rice response to cold stress.

### 2.4. Analysis of the Regulatory Elements of the OsSPX Promoter in DXWR and Their Correlations with Cold-Tolerance-Gene Expression

Cis-acting elements can be recognized from transcription factors and participate in the expression of tissue-specific and stress-response genes. In order to further study the roles and regulation mechanisms of *OsSPXs* in various biological processes, especially the response to cold stress, PLANTCARE and PlantTFDB were used to identify 2000 bp upstream sequences of four *OsSPXs* of DXWR. A total of 42 cis-acting elements were detected in PLANTCARE, of which 8 stress-response elements were the most, followed by light-response elements, growth- and development-related elements, and MYB transcription-factor-binding elements (Figure 4a). In general, *OsSPXs* are more likely to regulate stress responses, and the original elements of stress response are ARE, ABRE, STRE, WUN-motif, GC-motif, ABRE3a, LTR, and box-S. Hormone reaction elements include abscisic acid (AAGAA motif, AT-rich sequence, and ABRE4), gibberellin (GARE motif, GA motif, P-box, and TATC box), salicylic acid (TCA element), ethylene (ERE), and jasmonic acid (TGTCA motif and TGACG motif). In addition, the original elements of MYB, b-ZIP, and WRKY transcription-factor binding were also found in SPX. At the same time, 90 transcription-factor binding sites were identified with PlantTFDB, including ERF (22/90), WRKY (19/90), G2-like (10/90), Dof (9/90), NAC (4/90), MYB (4/90), and C2H2 (4/90) (Figure 4b). The diversity and function of these cis-acting elements and transcription-factor binding sites provide a deeper understanding of the biological functions of *OsSPXs*.

A Pearson correlation analysis showed that 36 cold-stress-response genes were positively correlated with the expression patterns of the four *OsSPXs* in different degrees (Figure 5). Among them, there were 24 genes related to the *OsSPX1* expression mode with rho ≥ 0.7, among which those with rho ≥ 0.9 were *OsMYB3R-2* (0.97), *OsMKK6* (0.96), *OsbZIP52* (0.93), *OsMST8* (0.93), *OsNAC5* (0.92), and *OsSPX4* (0.90). The expression patterns of 18 genes and *OsSPX2* with rho ≥ 0.7, specifically with and rho ≥ 0.9, were *OsNAC5* (0.95), *OsWRKY45* (0.91), and *OsGRF4* (0.90). There were 21 pairs of genes for *OsSPX4* with rho ≥ 0.7, and the correlations between *OsCDPK1* (0.99), *OsbZIP52* (0.95), *OsTCP21* (0.93), and *OsMYB3R-2* (0.92) were more than 0.9. The expression pattern of *OsSPX6* and five genes with rho ≥ 0.7, was *OsTPP1* (0.81). These results suggest that these *OsSPXs* may participate in the regulatory network of cold-tolerance genes, thus mediating the process of the rice cold-tolerance response.

## 3. Discussion

The SPX subfamily only contains the SPX domain. It has been widely studied because of its role in plant Pi signal transduction. Rice is one of the most important food crops in the world. Studying the genes in the process of stress resistance is of great significance for stabilizing rice yield. However, at present, members of the SPX subfamily, as regulatory proteins in the process of adversity, have received little attention. This study identified six *OsSPXs* from DXWR and tried to explore their roles in cold stress.

Similar protein structures may indicate that genes have similar functions [39,40]. By comparing the homology of proteins, we found that OsSPX1 and OsSPX2 are closely related, and OsSPX3 has high homology with OsSPX5 and OsSPX6, whereas OsSPX4 is distantly related to other OsSPXs. These findings were further confirmed by analyzing the motif composition of OsSPXs. OsSPX1 and OsSPX2 contain specific motif 7, whereas OsSPX3 and OsSPX5 have specific motif 8. OsSPX3 and OsSPX6 have conservative motifs 7 and 10. OsSPX4 does not contain motifs 6-10. The subsequent intragenomic collinearity analysis showed that *OsSPX1, OsSPX2*, *OsSPX3*, *OsSPX5*, and *OsSPX6* were derived from the replication of genomic fragments, which may come from the amplification of the whole genome. Homologous genes are highly conserved among species. Because they have similar important functions during evolution, they may be preserved, which has been found to be the case in gramineous crops [41]. Analysis of the collinearity between genomes revealed that *OsSPX1*, *4*, and *5* contained direct homologous genes in maize, DXWR, and wheat. Except for *OsSPX6*, other *OsSPXs* have homologous genes in the maize genome. Three maize SPX proteins (ZmSPX3, 5, and 6) can interact with ZmPHR1, with the exception that *ZmSPX5* and *6* can be induced by low phosphorus. The homologous genes of *ZmSPX6*, *5*, and *3* in DXWR are *OsSPX1*, *2*, and *4*, respectively. Interestingly, OsSPX1, 2, and 4 can also adjust the phosphate balance by combining their SPX domain with OsPHR2 [42,43]. This shows that SPX has a highly conservative function in maize and rice. Common wheat (AABBDD) underwent two crosses and genome doubling from diploid ancestors to form heterohexaploid crops. In theory, one rice gene should correspond to three homologous wheat genes [44,45]. For example, the homologous genes of OsSPX1 in wheat, *Traescs7a02g376200*, *Traescs7b02g277700*, and *Traescs7d02g372600,* were all induced due to Pi starvation. It is also interesting that *Traescs7a02g376200* and *Traescs7d02g372600*, as candidate genes for saline–alkali tolerance, were significantly upregulated after saline–alkali treatment [46]. However, we know that *OsSPX1* responds to Pi starvation and plays a positive regulatory role in cold tolerance. It can be seen that *OsSPX1* is highly conserved in rice and wheat and has similar functions in the process of evolution.

As a signal transduction protein, SPX plays a role in Pi signal transport. *OsSPX1*, *2*, and *4* are highly expressed throughout the rice growth period, indicating that they are important for rice growth. *OsSPX1* has been proven to be related to the cold tolerance of rice. Overexpression of *OsSPX1* can enhance the cold tolerance of transgenic rice and Arabidopsis. Through qPCR verification, four *OsSPXs* were significantly induced in the cold-tolerant material DXWR and the cold-sensitive material GZX49, which showed key roles in cold-tolerance processes. Importantly, the induction degrees in the cold-tolerant material DXWR of *OsSPX1* and *2* at 4 °C for 24 h and of *OsSPX4* and *6* at 4 °C for 4, 12, and 24 h, were significantly higher than that of the cold-sensitive material GZX49, and the expression levels increased with an increase in low-temperature treatment time. DXWR promoter analysis showed that *OsSPX1*, *2*, *4*, and *6* promoters contained cis-regulatory elements related to environmental stress and hormone induction, which may endow this gene with potential functions of responding to stress and regulating endogenous hormones. When plants are under external pressures, such as droughts, low or high temperatures, salt, and other stresses, these pressure signals activate corresponding transcription factors through a series of signals and combine with corresponding homeopathic regulatory elements, thus activating the expression of related genes to respond to the external signal stress. *OsSPX1*, *2*, *4*, and *6* all contain ARE, ABRE, MYB, jasmonic acid (TGTCA-motif and TGACG-motif), and AAGAA-motif, indicating that these genes may play a key role in improving the resistance of DXWR. In addition, we also identified a large number of stress-related transcription-factor binding sites: ERF, WRKY, G2-like, Dof, NAC, MYB, and C2H2. The existence of these transcription-factor binding sites confirmed the response of the OsSPX genes to stress. For example, OsSPX1 has 16 ERF binding sites and 5 WRKY.

Four *OsSPXs* and cold-tolerant genes have highly similar expression patterns under cold stress. For example, *OsMYB3R-2* is highly correlated with *OsSPX1* and *OsSPX4* expressions. Previous studies have shown that *OsMYB3R-2* is the main switch of tolerance to stress, which can improve the tolerance to stress by activating the expression of cold-response genes such as *OsDREB2A* and *O_S_DREB1* [47,48]. Interestingly, we also found MYB transcription-factor binding sites in the *OsSPX1* and *OsSPX4* promoter regions. To summarize, our study revealed the expression patterns of *OsSPXs* in different tissues and under cold stress, which highlighted the potential role of DXWR *OsSPXs* in cold-stress tolerance.

## 4. Materials and Methods

### 4.1. Plant Materials and Evaluation of Cold Tolerance at the Seedling Stage

In this study, Dongxiang wild rice (DXWR) and indica rice (GZX49) were used as materials. These rice materials were planted in the rice base of the Rice Institute of the Jiangxi Academy of Agricultural Sciences. The harvested seeds were incubated at 40 °C for about 36 h to break dormancy, and then soaked in deionized water at 30 °C for about 60 h to germinate. A total of 30 germinated seeds were selected from each strain and planted in the soil in the pot for 12 h light (28 °C) and 12 h dark (26 °C) cycles, with a humidity of 80%. When most seedlings had grown to trefoil shape, the weak seedlings were removed, and the rice materials used for low-temperature stress were treated at 4 °C for 0, 4, 12, and 24 h. The collected samples were stored at −80 °C. The rest were treated in the incubator at a low temperature of 4 °C for 84 h, and then resumed growth at 28 °C for 7 days. After 14 days of treatment, the survival rate was determined by calculating the living plants (leaves containing about 40% of green parts) and dead plants. This process was repeated three times independently for the two materials.

### 4.2. Identification of SPX Gene in DXWR

The sequence information of DXWR comes from a gene-annotation website (http://www.ricegermplasmgenome.com/, accessed on 26 March 2023), and a hidden Markov model containing the SPX-domain gene (PF03105) was used to search and screen the candidate gene in hmmer3 [49,50]. Then, it passed the NCBI conservative domain database (CDD; https://www.ncbi.nlm.nih.gov/cdd, accessed on 26 March 2023) and SMART database (http://smart.embl-heidelberg.de/, accessed on 12 March 2023) to confirm the integrity of the protein domain and that it only contained the SPX domain [51,52].

### 4.3. Analysis of Main Characteristics of SPX Gene in DXWR

Using MEGA11, the phylogenetic tree of DXWR was repeatedly constructed through neighbor connection (NJ), with guidance set to 1000 [53]. The distribution of all DUF26-domain genes on the DXWR chromosome was analyzed and visualized using TBtools [54]. The characteristic motif of the SPX gene was determined with MEME (http://meme-suite.org/tools/meme, accessed on 12 March 2023). The base sequence number was 10, and the site distribution order of each sequence was either 0 or 1 [55].

### 4.4. Repetitive Events and Collinearity Analysis of OsSPXs

The multilinear analysis tool MCscanX analyzes the replication events of genes in the genome and uses it for visualization through Circos [56]. Genome data of wheat (Triticum_aesivum. IWGSC.52) and maize (Zea_mys. Zm-B73-REFERENCE-NAM-5.0.52) were acquired from the Ensembl plant genome website (https://plants.ensembl.org/index.html, accessed on 20 December 2022). The collinear relationships between DXWR and maize and wheat were analyzed and visualized using JCVI [57].

### 4.5. Promoter Analysis

According to the position of the gene on the chromosome, we extracted the DNA sequence 2 kb upstream of the promoter of the OsSPX gene coding region of Dongxiang wild rice. Using PlantCARE (http://bioinformatics.psb.ugent.be/webtools/plantcare/html/, accessed on 26 March 2022), the cis-acting element was predicted [58]. At the same time, the cis-acting element was submitted to the PlantTFDB database (http://planttfdb.gao-lab.org/, accessed on 26 March 2022). Transcription factor binding sites at 2 kb upstream of the candidate OsSPX promoter sequence were also predicted [59].

### 4.6. Correlation Analysis of the OsSPX Gene and Cold-Tolerance Genes

Correlation network analysis between the OsSPX gene and cold-tolerance genes was performed using OmicStudio tools (https://www.omicstudio.cn/tool accessed on 26 March 2023) (R version 3.6.3 igraph1.2.6).

### 4.7. Quantitative PCR

Total RNAs were isolated with a TRIzol kit (Invitrogen, Carlsbad, CA, USA) according to the manufacturer’s instructions. The RNA was treated with DNase I (Invitrogen), and approximately 3 µg of total RNA was used to synthesize first-strand cDNA using oligo(dT)_18_ as a primer (Promega, Shanghai, China). The trans-intron ACTIN primer (ACTIN-M) was used to detect whether the reverse-transcribed cDNA still had genomic DNA, and the cDNA without genomic DNA was used for subsequent qPCR. Quantitative PCR was performed using gene-specific primers and the FastStart Universal SYBR Green Master (Roche) on a qPCR ViiA7 system (Applied Biosystems). Genes *ubiquitin* (*LOC_Os03g13170*) and *actin1* (*LOC_Os03g50885*), which were not differentially expressed in RNAseq, were used as the internal controls. The relative quantification method (2^−∆∆CT^) was used to evaluate gene expression levels [60]. As similar expression results were observed regardless of which control genes were used, the ubiquitin of expressions was applied for the relative expression analyses for every assayed sample. At least three biological replicates, each containing four that were technical, were performed for each experiment.

The primers were designed according to the DXWR reference genome using Primer3 [61] (http://redb.ncpgr.cn/modules/redbtools/primer3.php, accessed on 8 March 2022). The sequences were analyzed using Sequencer 5.0 (Gene Codes Corporation, Ann Arbor, MI, USA). All primers were synthesized at Sangon Biotech (Shanghai, China) and are listed in Appendix A.

## 5. Conclusions

In this study, six *OsSPXs* were identified in the reference genome of DXWR, and the conserved domain, collinearity, expression pattern, and cis-elements were analyzed. Expression analysis showed that *OsSPX1-6* had different expression levels in all tissues, indicating that these *OsSPXs* played an important role in rice growth and development. In addition, we also analyzed the expression of *OsSPXs* under low-temperature stress and found that *OsSPX1*, *2*, *4*, and *6* were strongly induced by low temperatures, and the expression level in DXWR was significantly higher than that in low-temperature-sensitive materials. Finally, the coexpression network of *OsSPXs* and cold-signal-related genes was constructed, indicating that *OsSPXs* may participate in the cold-tolerance process by participating in the known cold-stress-response gene pathway. In future research, these candidate genes should be knocked-out or over-expressed in rice for further functional verification. The excellent alleles of DXWR were polymerized into conventional rice through gene polymerization to enhance its cold tolerance under the condition of ensuring yield. In conclusion, our research results are valuable and will be useful for further explaining the biological function of *OsSPXs* and for genetic improvement.

## 6. Patents

This section is not mandatory but may be added if there are patents resulting from the work reported in this manuscript.

## Figures and Tables

**Figure 1 ijms-24-08755-f001:**
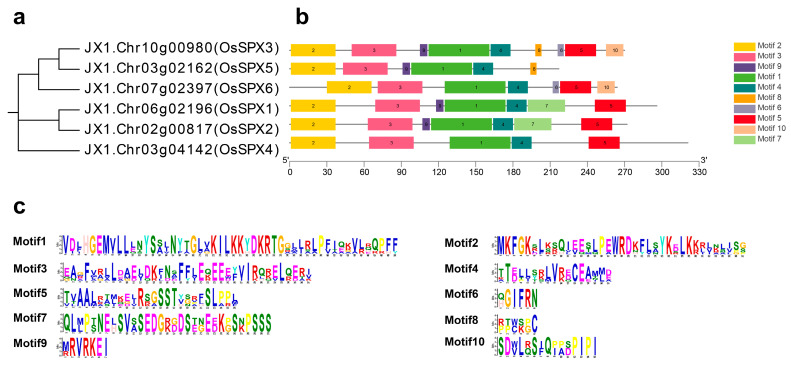
Phylogenetic relationship, domain, and conserved motif analysis of OsSPXs in DXWR. (**a**) The phylogenetic tree was constructed based on the full-length sequences of DXWR DUF26-domain-containing proteins using MEGA11. (**b**) Motif distribution of the OsSPXs. The OsSPX conserved motifs were determined with MEME (http://meme-suite.org/tools/meme, accessed on 26 March 2022) and visualized using TBtools. (**c**) Sequence logo of the OsSPX protein motifs. The height of each amino acid represents the relative frequency of the amino acid at that position.

**Figure 2 ijms-24-08755-f002:**
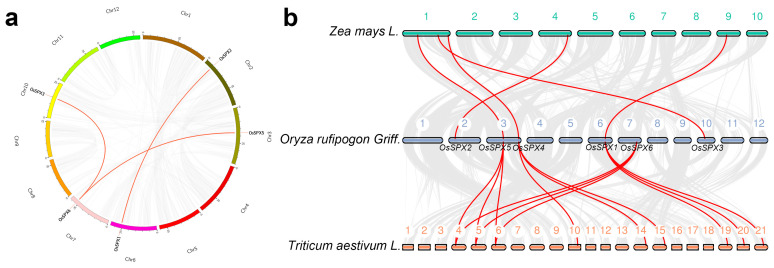
Collinearity relationship within the genome of *OsSPXs* of Dongxiang wild rice and its analysis with Gramineae plants. (**a**) The gray area shows all the collinear regions in the *OsSPXs* of DXWR, and the red line shows the collinear *OsSPX* pair. The position of the gene on the chromosome is shown at the top of each chromosome. (**b**) The gray lines in the background show the collinear region in the genome of DXWR and other plants. The colored-line region highlights the *OsSPX* pair that is collinear with different species.

**Figure 3 ijms-24-08755-f003:**
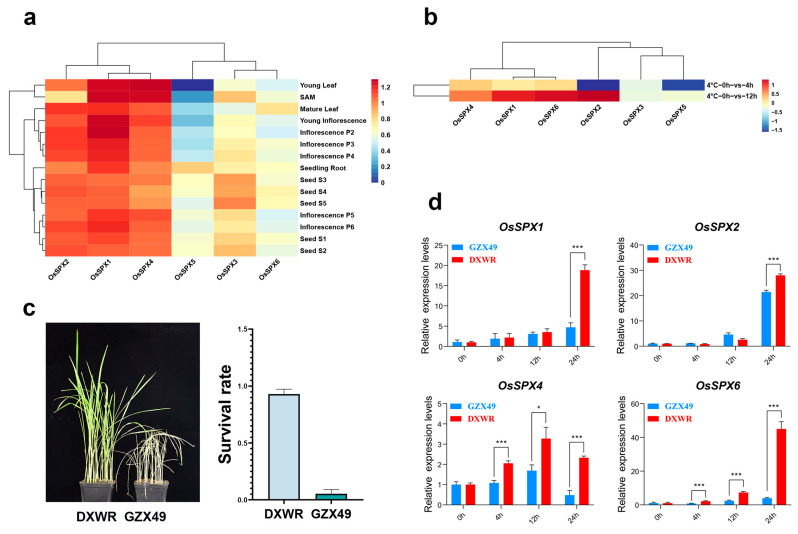
Expression patterns of *OsSPXs* in different rice tissues and their responses to cold stress. (**a**) The expression patterns of *OsSPXs* in five different growth stages (root, stem, leaf, flower, and seed). Heatmap was generated by taking the log2 fold of the FPKM ratio. (**b**) The expression mode of *OsSPXs* under treatment at 4 °C. Heatmap was generated by taking the log2 fold of the ratio. (**c**) The phenotypes of cold-tolerant rice and cold-sensitive rice after treatment at 4 °C and their survival rates. (**d**) The expression levels of *OsSPX1*, *2*, *4*, and *6* in GZX49 and DXWR after cold stress, detected using qPCR. The results were statistically analyzed using Student’s *t*-test (* *p* < 0.05, *** *p* < 0.005). Transcription levels relative to 0 h, which was set to 1, are presented as the mean and SE of triplicates. *LOC_Os03g13170* (Ubiquitin) is the control gene.

**Figure 4 ijms-24-08755-f004:**
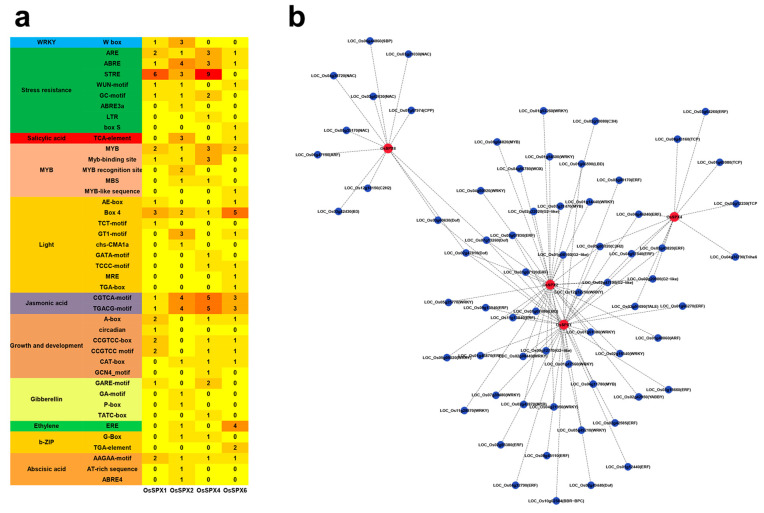
Analysis of the regulatory elements of SPX gene promoters in DXWR and their correlations with cold-tolerance-gene expression. (**a**) The cis-acting regulatory element (CRE) of DXWR *OsSPXs* were predicted using PlantCARE (http://bioinformatics.psb.ugent.be/webtools/plantcare/html/, accessed on 27 March 2022). (**b**) *OsSPXs* transcription-factor (TF) binding sites were determined using the PlantTFDB database (http://planttfdb.gao-lab.org/).

**Figure 5 ijms-24-08755-f005:**
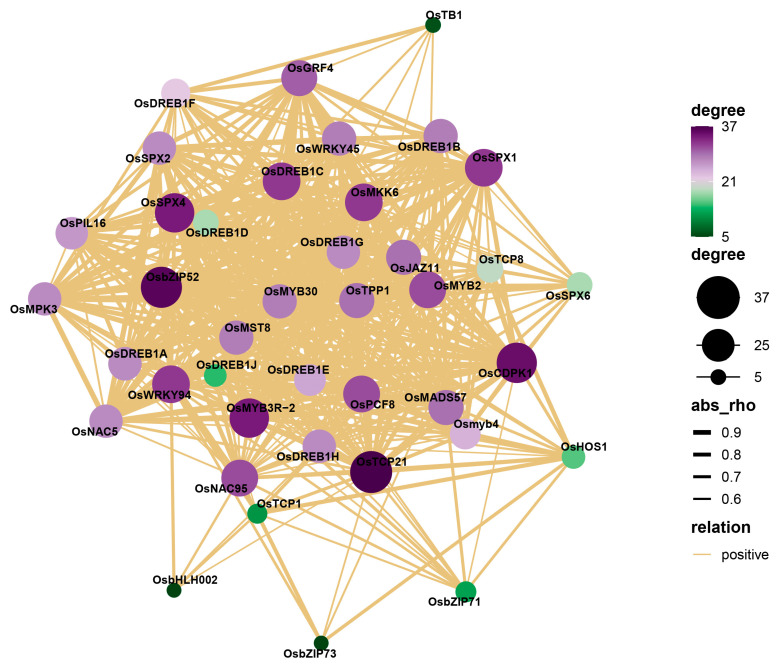
Correlation analysis of *OsSPXs* and cold-tolerance genes in DXWR (the size of the point and the thickness of the line indicates the degree of correlation, and the correlation is expressed by rho).

## Data Availability

Not applicable.

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
