# Peer review of "Genome-Wide Identification and Analysis of OsSPXs Revealed Its Genetic Influence on Cold Tolerance of Dongxiang Wild Rice (DXWR)"

_ijms, 2023, doi:10.3390/ijms24108755_

Round 1

Reviewer 1 Report

Low inorganic phosphate (Pi) is reportedly important in triggering cold acclimatization (Zhao et al., Plant biotechnology journal. 2009). In addition, OsSPX1 may play an important role in linking cold stress and Pi starvation signal transduction pathways in Arabidopsis and tobacco. Unfortunately, the authors do not refer to this article in their manuscript. However, the authors for the first time in rice plants (DXWR) identified 6 OsSPXs, the conserved domain, collinearity, expression pattern and cis-elements were analyzed and proved the participation of OsSPXs in the formation of cold tolerance.

The plan of experimental work is built correctly, logically correct, the conclusions reflect the results obtained.

Author Response

We gratefully appreciate for your valuable suggestion. OsSPX1 may play an important role in linking cold stress and Pi starvation signal transduction pathways in Arabidopsis and tobacco (Zhao et al., Plant biotechnology journal. 2009). Thank you very much for your suggestion. We have carefully reviewed the references and cited this article.

Reviewer 2 Report

The Genome-wide identification and analysis of OsSPXs manuscript revealed its genetic influence on cold tolerance of Dongxiang wild rice (DXWR) by Cheng Huang, Jilin Wang, Dianwen Wang, Jingjing Chang, Hongping Chen, Dazhou Chen, Wei Deng and Chunjie Tian presents interesting results complete research. well planned and executed.

However, a number of problems with terminology, typical for such works, greatly reduces the value of the results.

Actually, the publication of this generally good article in this form is simply unacceptable.

The reason is simple, fabric is a very specific concept. The study of tissue-specific expression is a very time-consuming technique, accompanied by many of its own problems and artifacts. The authors of the article, like many authors of molecular biological works, do not understand the difference between an organ and a tissue. I will not go into too much detail, but it will not be difficult for the authors to familiarize themselves with the fact that there are relatively few tissues in the leaf: lower epidermis, upper epidermis, parenchyma, protophloem and protoxylem tissues, vascular bundle parenchyma, often there are also stomatal cells and trichomes. These tissues often have different ploidy and under different conditions can have different relationships with each other, which makes this work difficult... within the framework of double fertilization and containing the genome of both the mother plant and the paternal plant in the nucleus, but not in the same ratio as in other tissues). Such things should be obvious to the authors and should not raise questions from readers.

For this reason, in all cases it is necessary to correct tissues for organs!!!

In the same connection, the use of just one gene as a housekeeping gene should be carefully weighed, especially in the specific conditions that the authors investigated, and the validity of their choice should be discussed in detail.

This made the work less significant and reliable and should be taken into account in the future.

It is also desirable for the authors to add to the discussion, the features of the reaction of cells to this effect. Since I could not find this section, which would enrich the discussion.

The fact is that at a temperature of about 4 degrees, the highest density of water is observed, at which gases are removed from the water and breathing processes slow down, plants are able to maintain a certain range of temperature transition, but not as sharp as the authors indicated. It is well known that the very transition of temperatures with a jump above 10 degrees is a critical damage for many plants and leads to an irreversible change in membranes.

Figure 4C is of some interest and is not readable as presented. Or do it separately by zooming in, or bring it down to the application and highlight the lines with different colors, is zooming necessary, since this is the main result of your analysis?

This study, of course, gives a general idea of the processes of transformation of work, but without taking into account physiological parameters, it looks less informative. I recommend the authors to revise the manuscript taking into account the comments. And avoid such mistakes in the future.

Author Response

We gratefully appreciate for your valuable suggestion. Dongxiang wild rice is the highest latitude wild rice in the world, with strong cold tolerance. It is widely believed that indica rice originated in Southeast Asia and is very sensitive to low temperatures. At the same time, in the long process of rice breeding, the excellent cold tolerance genes of wild rice were lost in the process of artificial and natural environment selection. Our research team has been committed to mining the stress tolerance genes of Dongxiang wild rice for a long time, and hopes to improve the cold tolerance of rice in the future to adapt to the occurrence of extreme weather. We treated at 4 ° C for 84 hours and recovered at 28 ° C for 7 days, and found that the survival rate of Dongxiang wild rice was greater than 90%, while the survival rate of cold sensitive indica rice(GZX49) was around 10%. Under these treatment conditions, we found that it was possible to distinguish between cold tolerant and non cold tolerant materials to the greatest extent possible(This experiment was repeatedly validated and previously published in the journal Plants). The reason for choosing the SPX gene family for analysis is that, based on previous research results, overexpression of OsSPX1 can significantly improve the cold tolerance of rice, Arabidopsis, and tobacco. At the same time, we also identified the differences between members of the OsSPXs in cold tolerant and intolerant materials through GWAS (this part has not been published). We believe that the natural excellent allelic variation of Dongxiang wild rice is the main reason for giving strong cold tolerance to rice. Based on this, we have created a near isogenic line between Dongxiang wild rice and indica rice material (GZX49) to screen for differences in genetic background as much as possible, identify the cold tolerance genes of Dongxiang wild rice from gene fragments, and have the potential to be used in rice breeding in the future (this part of the content has not been published). It is generally believed that different expression patterns of genes in different tissue parts may indicate their specific functions. High expression genes throughout the entire growth period may perform important biological life activities, while genes expressed in specific tissues may participate in specific growth processes. In addition, the modifications you mentioned regarding the images have been improved in the newly submitted version.

Round 2

Reviewer 2 Report

Dear authors, it was interesting for me to read about your plans, but still I ask you to read the review and follow what is written there or contact plant anatomy specialists, as I repeat: the leaf is not a tissue, but an organ! like "seeds" (which rice does not have, since they have a fruit that is three different genomes - shells - contain the maternal genome, the endosperm is a triploid genome of maternal and paternal origin, see what double fertilization is, and the embryo is genome of a new hybrid type). Thus, the kernels that you call seeds are not tissue, but a separate new sporophyte. Without understanding such things, it is extremely difficult to work with plants and you should correct these mistakes.

Also, the magnification did not lead to the fact that anyone can evaluate the result in Figure 3C. In order for the lines to be visible, you should use the mode with the edges of the lines highlighted in a different color, or use a different way of presenting this material. Consult with a design specialist.

I repeat, the concept of tissue, tissue-specific, etc. not applicable for your samples and your data.

Author Response

Various rice tissues/organs (seedling root, mature leaf and Y leaf) and stages of reproductive (panicle and seed) development were used for RNA extraction and hybridization on Affymetrix microarrays. Different stages of panicle and seed development have been categorized according to panicle length and days after pollination, respectively, based on landmark developmental events as follows: up to 0.5 mm, shoot apical meristem and rachis meristem (SAM); 0-3 cm, floral transition and floral organ development (P1); 3-10 cm, meiotic stage (P2 and P3); 10-15 cm, young microspore stage (P4); 15-22 cm, vacuolated pollen stage (P5); 22-30 cm, mature pollen stage (P6); 0-2 dap, early globular embryo (S1); 3-4 dap, middle and late globular embryo (S2); 5-10 dap, embryo morphogenesis (S3); 11-20 dap, embryo maturation (S4); 21-29, dormancy and desiccation tolerance (S5). These stage specifications are approximations based on information from Itoh et al., 2005, Plant Cell Physiol, 46, 23-47.

Tissue is the basic functional unit of organs, and organs are composed of tissue. Thank you very much for your suggestion, which will help me re understand the concepts of tissue and organs and better understand rice in my future work. In addition, I believe that heat maps of expression patterns at different growth stages can help us analyze whether genes are expressed constitutively or tissue-specific, which provides a reference for us to determine the function of genes. For example, CTB4a is widely expressed in various tissues, especially in the tapetum and anther septal vascular bundles. It is found that it can improve the cold tolerance of rice at booting stage, improve the pollen fertility of rice under low temperature conditions, and thus improve the seed setting rate and yield (Zhang et al. 2017). In addition, according to your suggestion, I have made new modifications to Figure 3c.

Round 3

Reviewer 2 Report

I consider this work significant and valuable. I hope this experience will allow us to obtain new valuable results in the future, taking into account a new perspective.